# GPC1 promotes the growth and migration of colorectal cancer cells through regulating the TGF-β1/SMAD2 signaling pathway

**Fei Lu[1], Shuran Chen[1], Weijun Shi[1], Xu Su[2], Huazhang Wu[2]\*, Mulin Liu[1]\***

**1** Gastrointestinal Surgery, the First Affiliated Hospital of Bengbu Medical College, Bengbu, China, **2** School of Life Science, Anhui Province Key Laboratory of Translational Cancer Research, Bengbu Medical College, Bengbu, China

\* 1143274470@qq.com (HW); 2583375185@qq.com (ML)

**Data Availability Statement:** All relevant data are within the paper and its Supporting Information files.

## Abstract

In this study, we analyzed GPC family genes in colorectal cancer (CRC) and the possible mechanism of action of GPC1 in CRC. CRC patient data were extracted from The Cancer Genome Atlas, and the prognostic significance of GPC1 expression and its association with clinicopathological features were identified by Kolmogorov–Smirnov test. CRC patients with high GPC1 expression had poor overall survival compared with patients with low GPC1 expression. In vitro experiments demonstrated that knockdown of GPC1 significantly inhibited the proliferation and migration and promoted cell apoptosis in CRC cell lines. Gene Ontology analysis of differential genes indicated that GPC1 may influence the TGF-β1 signaling pathway. Additional experiments revealed that silencing GPC1 suppressed the levels of TGF-β1 and p-SMAD2 but increased the expression of SMAD2. Taken together, these findings suggest that GPC1 may function as a tumor promoter in CRC cells through promoting TGF-β signaling pathway. Our results also indicate that GPC1 may serve as a critical effector in CRC progression and a new potential target for CRC therapy.

## Introduction

Colorectal cancer (CRC) is one of the most common cancers in the United States, with an estimated 147,950 and 53,200 new cases and deaths each year, respectively [1]. In 2015, an estimated 376,300 cases and 150,000 deaths occurred in China [2]. Approximately 20% of patients with CRC have metastatic CRC [3], and 40% of CRC patients exhibit recurrence after treated localized disease [4]. The prognosis of metastatic CRC is poor, with a 5-year survival rate of less than 20% [3]. Therefore, the identification of new markers and novel treatment methods for CRC is critical.

Glypicans (GPCs) are a family of heparan sulfate proteoglycans (HSPGs) that interact with the plasma membrane through a glycosylphosphatidyl inositol anchor [5]. In humans, six GPC family members have been identified, including GPC1, GPC2, GPC3, GPC4, GPC5 and GPC6 [6]. Melo et al. studied GPC1 expression in the peripheral blood of 190 patients with pancreatic ductal adenocarcinoma and 100 healthy volunteers and found that GPC1 was

**Funding:** LF and LM both are funded by the following: Anhui Natural Science Foundation under Grand (No. 2108085MH291), and the 512 Talent Cultivation Plan for subject leader of Bengbu Medical College (by51201107). LF and WH both are funded by the following: Anhui Natural Science Foundation under Grand (1908085MH257). The funders had no role in study design, data collection and analysis, decision to publish, or preparation of the manuscript.

**Competing interests:** The authors declare no competing interests.

highly expressed in patients with cancer compared with the healthy volunteers; furthermore, larger tumors showed a higher positive rate of GPC1. Li et al. found that GPC1 regulates the occurrence of esophageal cancer through the PTEN/Akt/β-catenin signaling pathway [7]. In addition, the authors found that GPC1 can be used as a biomarker for the recurrence of stage III CRC, and GPC1 may be involved in EMT activation, invasion and migration of CRC cells [8]. Further studies revealed the increased expression of GPC1 in prostate cancer, endometrial cancer, lung cancer and other cancers [9–11]. However, the mechanism by which GPC1 regulates the occurrence and progression of CRC is still unclear.

In this study, we analyzed the influence of GPC family proteins in CRC and the possible mechanism of action. Our findings indicate that GPC1 expression was associated with prognosis in CRC patients and reveal that GPC1 promotes the TGF-β signaling pathway in CRC cells.

## Materials and methods

### Patients and specimens

Between January and February 2022, we collected 4 pairs of cancer and paracarcinoma tissues from patients with CRC at the Gastrointestinal Surgery Department of the First Affiliated Hospital of Bengbu Medical College. Our study was approved by the ethics committee of Bengbu Medical College. We obtained informed consent from every patient. All specimens were stored at −80˚C.

### Data source

The expression profiles and clinicopathological data of CRC and non-tumor tissues were obtained from The Cancer Genome Atlas (TCGA, https://portal.gdc.cancer.gov/) [12]. The inclusion criteria for clinical information were set as follows: (1) patients had complete clinical information; and (2) the follow-up time of samples exceeded 30 days.

### Analysis of TCGA mRNA profiles

We used the "limma" package to identify differentially expressed mRNAs between 473 CRC tissues and 41 non-tumor tissues from TCGA (|FDR|>1, $P<0.05$) [13]. The Kaplan–Meier method was used for univariate analysis, followed by log-rank test for assessing the differences of overall survival (OS) among different groups ($P<0.05$). Next, we assessed the correlation between GPC1 and clinicopathological information using Kolmogorov–Smirnov test ($P<0.05$). Finally, we identified the GPC1-related genes using Pearson's correlation test ($P<0.05$, $|R|>0.5$). The STRING database (https://string-db.org/), a publicly available comprehensive resource, was used to predict the relationships between target genes.

### Functional enrichment analysis

Gene Ontology (GO) enrichment analysis was performed using "clusterProfiler" package in R (4.1.0), and the "org.Hs.eg.db" package was used as the reference data. The P value was corrected by the Benjamin–Hochberg method, with a P value $<0.05$ and a q value $<0.05$ being the cut-off criteria.

### Immunohistochemistry

Colorectal cancer and paracarcinoma tissues were dewaxed in xylene and rehydrated through different gradients of alcohol. Then, we placed paraffin in 3% $H_2O_2$ for 15 min at 22˚C. Next, the slide was heated in citrate buffer. After washing several times with PBS (pH = 7.2), the slide was placed into the solution with a primary antibody against GPC1 (dilution 1:200,

16700-1-AP, Proteintech, China) at 4˚C for more than 12 h. Then, we added secondary antibody at 22˚C for 10 min. Finally, the slide was stained after the addition of 3,3′-diaminobenzidine (DAB) solution.

## Cell culture and reagents

HCT116, SW480 cell lines were obtained from Cell Bank, Type Culture Collection, Chinese Academy of Sciences (Shanghai, China). All cells were cultured in DMEM (Gibco, USA) supplemented with 10% fetal bovine serum (Sigma-Aldrich, Australia) and penicillin-streptomycin (Invitrogen, USA) in a 90%–95% humidified atmosphere of 5% CO2 at 37˚C.

## siRNA-mediated gene silencing

SW480 and HCT116 cells were seeded in 6-well plates ($2 \times 10^5$ cells/well) or 96-well plates ($2 \times 10^3$ cells/well) 24 h before transfection. Two siRNAs designed to silence GPC1 and a control siRNA were obtained from Genepharma (Shanghai, China). The siRNA sequences are as follows: Control siRNA: 5′–UUCUCCGAACGUGUCACGUTT–3′; siGPC1#1: 5′–CCUGGAUA GUUAAGGGCUUTT–3′; and siGPC1#2: 5′–CCUUUCUGCCUUUUAAUUUTT–3′. siRNAs were transiently transfected into cells with Lipofectamine 8000 transfection reagent (Beyotime, Shanghai, China) according to the manufacturer's instructions.

## Cell proliferation assay

The CCK-8 Cell Counting Kit (Beyotime) was used to assess the proliferation of HCT116 and SW480 cells. Transfected cells were seeded in 96-well plates at a density of $2 \times 10^4$ cells/mL (100 μL/well) and incubated at 37˚C with 5% $CO_2$. On days 0 (at 48 h after transfection), 1, 2 and 3, CCK-8 reagent (10 μL) was added to cells and cells were incubated for 4–6 h. The optical density of the solution (OD450) in each well was measured by spectrophotometry (BioTek).

## Apoptosis assay

The Annexin V-FITC Apoptosis Detection Kit (Beyotime, Shanghai, China) and a flow cytometer (Becton Dickinson, Franklin Lakes, NJ, USA) were used to evaluate the apoptosis of SW480 or HCT116 cells after transfection for 48 h. Briefly, cells were prepared into a 1×106 cells/mL cell suspension with 1 mL 1× Binding Buffer. Next, 200 μL of the cell suspension was mixed with 5 μL of Annexin V-FITC reagent in a centrifuge tube and the sample was mixed for 10 min in the dark at room temperature. PI reagent (5 μL) was then added to the centrifuge tube and samples were incubated in the dark at room temperature for 5 min. PBS was added to adjust the mixture to a final volume of 500 μL. After 30 min, the cells were evaluated by flow cytometry and the data were evaluated using CytoFLEX (Beckman Coulter) to calculate the number of apoptotic cells.

## Cell cycle analysis

At 48 h after transfection, SW480 and HCT116 cells ($2 \times 10^6$/ml) were collected, washed twice with PBS buffer and centrifuged at $1,000 \times g$ for 5 min at 4˚C. The supernatant was discarded and cells were fixed with ice-cold 70% ethanol for at 2 h at 4˚C. The cells were centrifuged at $1,000 \times g$ for 5 min at 4˚C, washed twice with PBS and centrifuged again at $1,000 \times g$ for 5 min at 4˚C. The supernatant was discarded and cells were incubated with 50 μl RNase I (1 μg/ml; Beyotime, Shanghai, China) at 37˚C for 1 h in the dark. Next, 200 μl propidium iodide (20 μg/ml) was added and the samples were held at room temperature for 20 min according to the manufacturer's protocol.

## Wound-healing assay

At 48 h after transfection, a sterilized 10-μL pipette tip was used to create a scratch in the monolayer of transfected CRC cells cultured in 6-well plates. After removing cell debris by washing with phosphate buffer saline (PBS), DMEM containing 2% FBS was added to each well and cells were cultured under 5% $CO_2$ and 37°C for 48 h. The scratch width was measured at 0 h and 48 h using ImageJ software and an inverted optical microscope. The relative scratch width was calculated as the ratio of the scratch width at 48 h to the width at 0 h.

## Transwell assay

At 24 h after transfection of CRC cells, 100 μL of cells ($1 \times 10^5$ cells) were plated in serum-free DMEM medium in the top of a transwell chamber, and 600 μL of 20% DMEM medium was included in the bottom chamber. After 72 h of incubation under 37°C and 5% $CO_2$, 99.99% methanol was used to fix the chamber, and 0.1% crystal violet was used for staining. A cotton swab was used to remove cells from the top chamber. Migrated cells were photographed and counted with an inverted optical microscope.

## Real-time quantitative polymerase chain reaction (RT-qPCR)

Total RNA was isolated from CRC cells using TRizol reagent (Invitrogen, USA) according to the manufacturer's instruction. Total RNAs were quantitatively analyzed and reverse-transcribed using a reverse transcription kit (PrimeScript RT Reagent Kit, RR047A, TaKaRa, Japan) to synthesize cDNAs. RT-qPCR was performed using a real-time quantitative PCR kit (A46113, Applied Biosystems, USA) on a QuantStudio 3 Real-Time PCR System (Thermo Fisher, USA). Each sample was run in triplicate. The sequences of primers used are listed in Table 1. The expression of target genes was calculated by the $2^{-\Delta\Delta Ct}$ method [10]. GAPDH mRNA was used as a housekeeping gene for normalization of gene expression.

## Western blotting

SW480, and HCT116 cells were lysed using RIPA buffer (Beyotime, Shanghai, China). Equal amounts of protein were separated by 10% SDS–PAGE and electrophoretically transferred to a PVDF membrane (Millipore, Billerica, MA, USA). The membrane was blocked with 5% non-fat milk for 2 h of blocking and then incubated with primary antibodies for 12 h at 4°C. After three washes, the membranes were incubated with secondary antibodies (1:5000) for 4 h followed by washing with TBST washing buffer. The protein bands were visualized using BeyoECL Star (Beyotime) and BIO-RAD Gel Doc XR+ (USA) and Image J software were used to analyze protein bands. The primary antibodies used are as follows: primary antibodies against GPC1 (1:1000, 16700-1-AP, Proteintech); TGF-β1 (1:1000, 21898-1-AP, Proteintech); SMAD2 (1:1000, 12570-1-AP, Proteintech); and p-SMAD2 (1:1000, ABP50459, abbkine).

**Table 1. Sequences of primers for qRT-PCR.**

| ID | Forward sequence(5′-3′) | Reverse sequence(5′-3′) |
|---|---|---|
| **GPC1** | TGAAGCTGGTCTACTGTGCTC | CCCAGAACTTGTCGGTGATGA |
| **GAPDH** | AGATCCCTCCAAAATCAAGTGG | GGCAGAGATGATGACCCTTTT |
| **TGF-β1** | CTAATGGTGGAAACCCACAACG | TATCGCCAGGAATTGTTGCTG |
| **SMAD2** | TCATAGCTTGGATTTACAGCCAG | TTCTACCGTGGCATTTCGGTT |
| **SMAD3** | TGGACGCAGGTTCTCCAAAC | CCGGCTCGCAGTAGGTAAC |

## Statistical analysis

Log-rank test was used to compare the Kaplan–Meier survival curves between high and low expression of GPC1, and the median was used as cutoff. $P < 0.05$ indicated statistical significance.

## Results

### Elevated expression of GPC1 is positively associated with the survival, disease stage and TNM stage of CRC patients

We analyzed the mRNA expression of GPCs in CRC and non-tumor tissues from TCGA database using the "limma" and "survival" program packages of R software. The results showed that the expressions of GPC1 and GPC2 were higher ($P < 0.001$) in CRC tissues than non-tumor tissues (Fig 1A). Furthermore, Kaplan–Meier analysis revealed that high expression of GPC1 and GPC2 were associated with shorter ($P < 0.01$) overall survival in CRC patients (log-rank test) (Fig 1B). We also found that GPC1 expression was lower ($P < 0.01$) in early stage tumors (stage I or II, T1–2, N0 or N1, M0) than in late stage tumors (stage III or IV, T3–4, N2, M1) (Fig 2A–2E) (S1 Table). GPC2 expression was lower ($P < 0.05$) in early stage tumors (stage I or II, T1–2, N0 or N1) than in late stage tumors (stage III or IV, T3–4, N2). There were significant differences of GPC2 expression in M0 and M1 stage ($P > 0.05$) (Fig 2F–2J and S1 Table). In addition, compared with that in para-carcinoma tissues, the protein expression of GPC1 in tumour tissues was elevated (Fig 2K). These results indicate that CRC patients with a high expression of GPC1 had a poor prognosis. Furthermore, GPC1 may function as an oncogene in the tumorigenesis and development of CRC and may also serve as a potential molecular marker for diagnosis and prognosis prediction for CRC.

### SiRNA-mediated GPC1 gene silencing significantly inhibits cell growth, induces cell cycle arrest, and promotes apoptosis of CRC cells

To explore the role of GPC1 in CRC cells, we performed siRNA-mediated gene silencing of GPC1 (si-GPC1-1 and si-GPC1-2). qRT-PCR and western blot analysis were conducted to measure GPC1 mRNA and protein expression in HCT116 and SW480 cell lines at 48 and 72 h after transfection. GPC1 mRNA expression levels were knocked down by up to 60% with si-GPC-1-1 and si-GPC1-2. We then evaluated the effects of GPC1 silencing on cell proliferation using the CCK-8 assay. The results showed that silencing GPC1 significantly ($P < 0.05$) inhibited the proliferation of HCT116 and SW480 CRC cells compared with cells transfected with control siRNA (Fig 3A–3D).

To further study the biological function of GPC1 in the development of CRC, we evaluated the effect of GPC1 silencing on the cell cycle and apoptosis of CRC cells by flow cytometry assay. Knockdown of GPC1 significantly ($P < 0.05$) induced apoptosis and cell cycle arrest at the S phase in SW480 and HCT116 cells (Fig 4A–4D). These results indicated that silencing GPC1 inhibits cell growth, induces S phase arrest and promotes apoptosis of CRC cells.

### Silencing GPC1 significantly reduces cell migration ability in CRC cells in vitro

To further investigate the role of GPC1 in CRC, cell motility was assessed by wound healing and transwell migration assays. In wound healing assays, cells with GPC1 knockdown showed reduced migration compared with control cells in SW480 and HCT116 cell lines (Fig 5A and

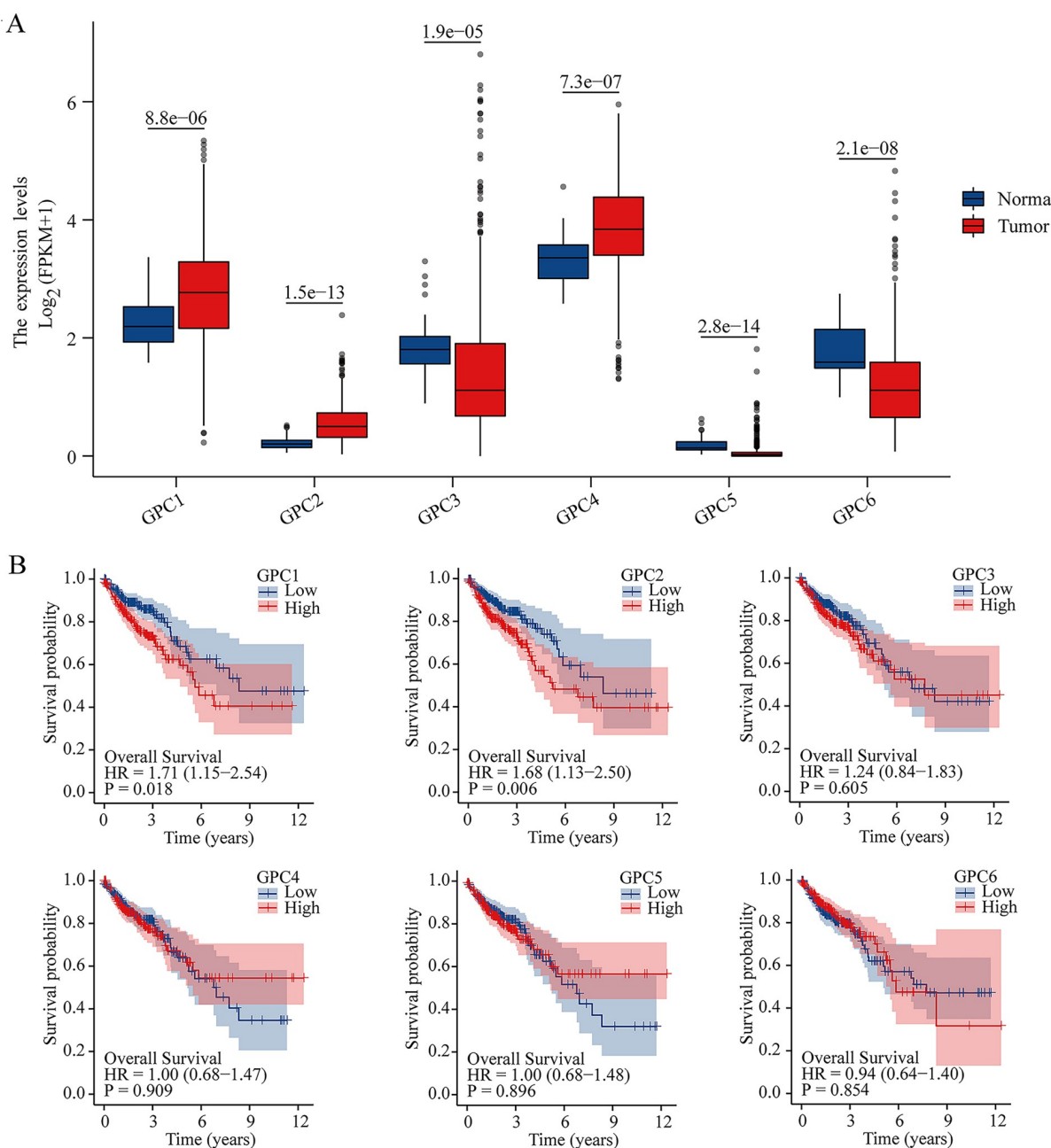

**Fig 1. Differential expression of GPC family genes in colorectal cancer (CRC) patients.** (A). Gene expression of six GPC family members in cancer tissues (473 samples) and non-tumor tissues (41 samples) in CRC patients (TCGA) using Student's t test. Blue indicates non-tumor tissues, and red indicates CRC tissues. (B). Kaplan–Meier analysis revealed that CRC patients (453 samples, TCGA) with high GPC1 expression (225 samples, TCGA) and high GPC2 expression (225 samples, TCGA) had a poor prognosis compared with patients with low GPC1 and GPC2 expression.

5B). Similarly, transwell migration assay indicated that GPC1 knockdown significantly (P<0.01) impaired cell migration ability in SW480 and HCT116 cells (Fig 5C and 5D). These results showed that silencing GPC1 impairs cell migration activity in CRC cells, further implying an important role for GPC1 in the progression of CRC.

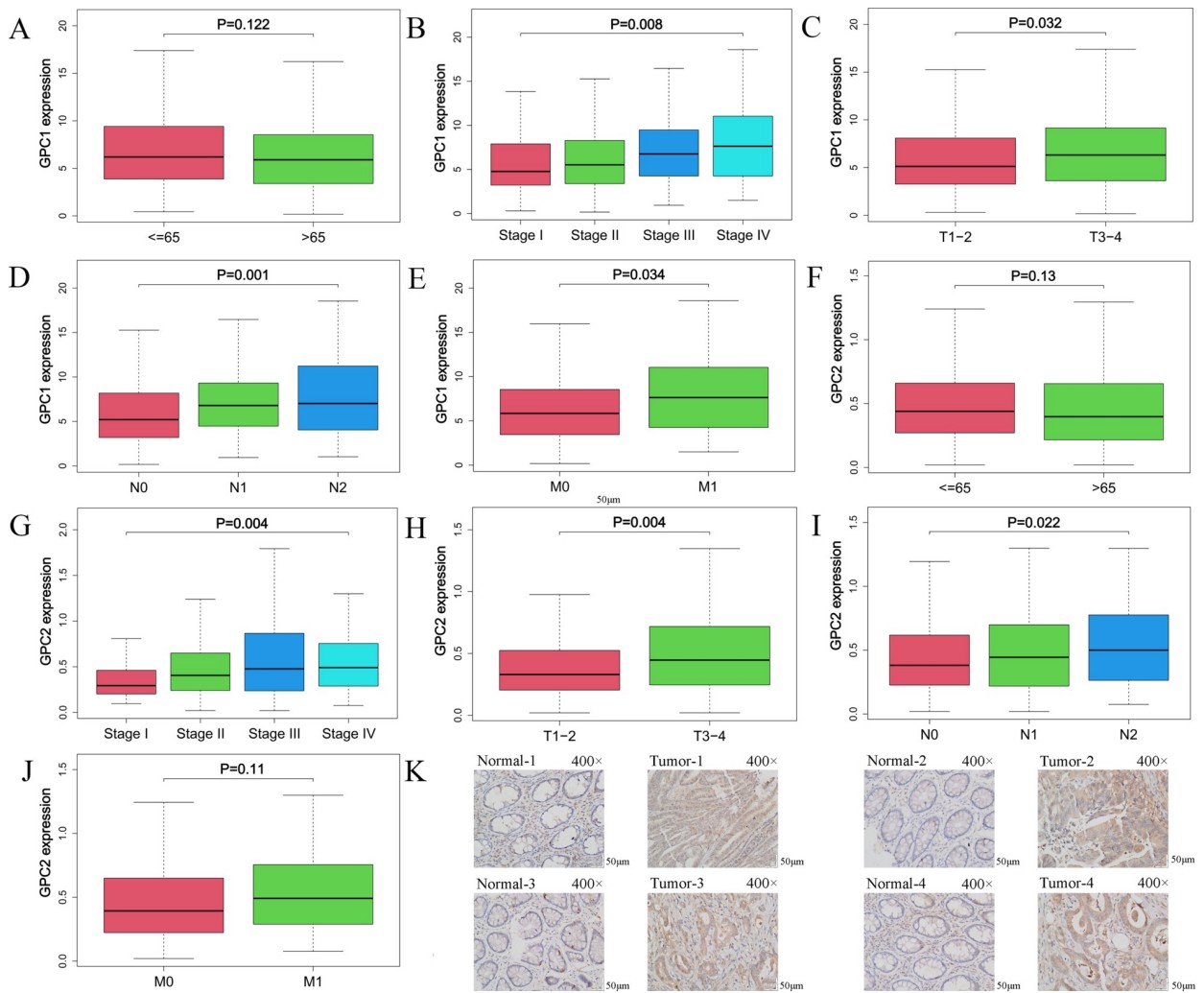

**Fig 2. Elevated expression of GPC1 is associated with clinicopathological features of CRC.** (A–J). GPC1 and GPC2 mRNA expression analysis in TCGA CRC samples (453 samples) by age, stage, T, N and M using Kolmogorov–Smirnov test. (K). Immunohistochemical staining for GPC1 in CRC tissues and normal tissues.

## GPC1 regulates the TGF-β1/SMAD2 signaling pathway

To explore the mechanism underlying the involvement of GPC1 in CRC, we first analyzed the possible signaling pathways mediated by GPC1 by bioinformatics assay using "limma" bioconductor to identify the RNAs related to GPC1. The GPC1-related RNAs are shown in a heatmap in Fig 6A. Next, we used GO enrichment analysis on the target genes that were screened (Fig 6B). We then predicted the relationships between GPC1-related target genes (Fig 6C). Two different bioinformatics analyses indicated that GPC1 may regulate the TGF-β1 signaling pathway in CRC cells (Fig 6B and 6C), which is consistent with our previous findings [14]. Notably, previous studies showed that the TGF-β signaling pathway is involved in the occurrence and development of CRC [15].

To confirm the results of the bioinformatics analysis, we explored the effect of GPC1 silencing on the expression of TGF-β1 signaling pathway–related molecules such as SMAD1 and SMAD2 by qRT-PCR. TGF-β1 and SMAD2 mRNAs were significantly down-regulated and up-regulated, respectively, in HCT116 and SW480 cells after the silencing of GPC1 expression

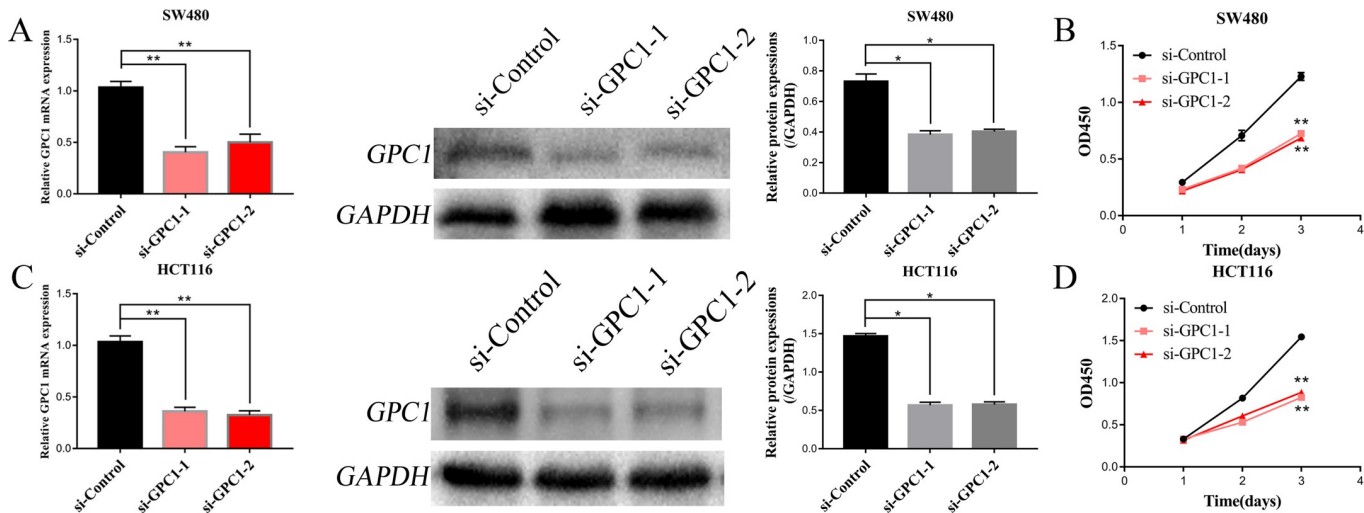

**Fig 3. Silencing GPC1 inhibits CRC cell proliferation.** (A, C). CRC cell lines were transfected with siRNA targeting GPC1, and silencing efficiency at the mRNA and protein levels was determined. (B, D). Cell proliferation was inhibited in SW480 and HCT116 cells by silencing GPC1. * $p < 0.05$, ** $p < 0.01$.

(Fig 6D and 6F). We then evaluated the effects of GPC1 silencing on the protein levels of TGF-β1, SMAD2 and p-SMAD2 by western blot assay. The results showed that the levels of TGF-β1 and p-SMAD2 proteins were significantly down-regulated, and the expression of SMAD2 protein was significantly up-regulated in HCT116 and SW480 cells following the silencing of GPC1 expression (Fig 6E and 6G). Together these results suggest that GPC1 may be involved in the proliferation, apoptosis and migration of CRC cells through regulating the TGF-β1/SMAD2 signaling pathway.

## Discussion

With the development of bioinformatics, several key genes that play an important role in the occurrence and development of CRC have been identified [16]. In this study, we analyzed the gene expression of the GPC family in CRC. GPCs belong to the HSPG family and interact with a variety of protein ligands, proteases, cytokines, chemokines, adhesion molecules and ECM proteins, thereby producing a variety of structures and signal transduction functions [17]. HSPG proteins exhibit functions in a wide range of biological processes, such as development, hemostasis control, and inflammation, and are associated with cell survival [18]. In addition, HSPG proteins are involved in cell adhesion, movement, proliferation, differentiation and apoptosis [14] and thus have been linked to the enhancement of malignant diseases.

In this study, we found that high expression of GPC1 was significantly related to the poor prognosis of patients with CRC. This finding is consistent with previous reports [19]. Several studies demonstrated that GPC1 is involved in the formation and development of different types of tumors [20, 21]. Experimental studies revealed that the expression of GPC1 was significantly increased in the extracellular vesicles released by the mouse MC38 CRC cell line [22]. In addition, TMT-MS methods to study crEV in patients with CRC found that GPC1 can be used as a biomarker for the detection of early CRC [23].

A previous study using bioinformatics analysis showed that GPC1 may participate in the occurrence and development of CRC through the glycolytic pathway [23]. Other analyses found that GPC1 may be involved in CRC development through influencing intestinal tumor hypoxia on immune cells in the tumor microenvironment [24]. However, these findings have not been verified by experiments and the precise mechanism by which GPC1 participates in

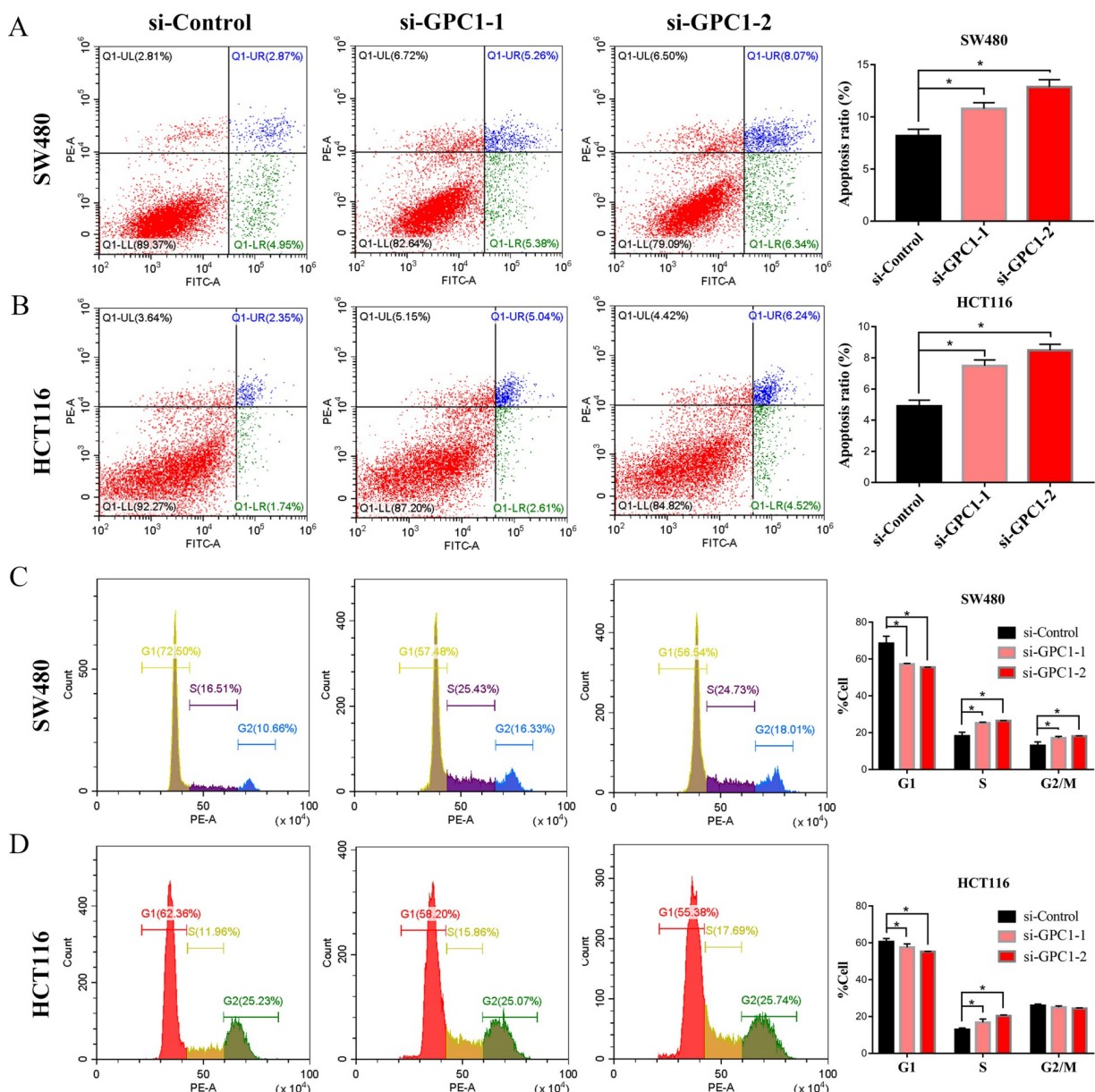

**Fig 4. Knockdown of GPC1 induces apoptosis and S phase cell cycle arrest in CRC cells.** (A, B). GPC1 silencing promoted SW480 and HCT116 cell apoptosis. (C, D). Knockdown of GPC1 induced S cell cycle arrest in SW480 and HCT116 cells. * p<0.05.

the regulation of CRC is still unclear. In this study, our in vitro experiments revealed that GPC1 gene silencing in the HCT116 and SW480 CRC cell lines resulted in inhibition of cell proliferation, S phase cycle arrest, induction of apoptosis and reduced migration of CRC cells. We further showed that GPC1 may be involved in the occurrence and development of CRC by activating the TGF-β/SMAD2 signaling pathway. Decreased GPC1 expression suppresses pancreatic cancer cell growth by modifying TGF-β signaling [25]. GPC1 has been shown to interact with TGF-β and its receptors to stabilize their assembly for enhanced Smad signaling. Downregulation of GPC1 expression resulted in a slightly altered response toward TGF-β1, activin-A, and BMP2 in terms of growth, p21 induction, and Smad2 phosphorylation, ultimately leading to decreased anchorage-independent growth of T3M4 and PANC-1 cells.

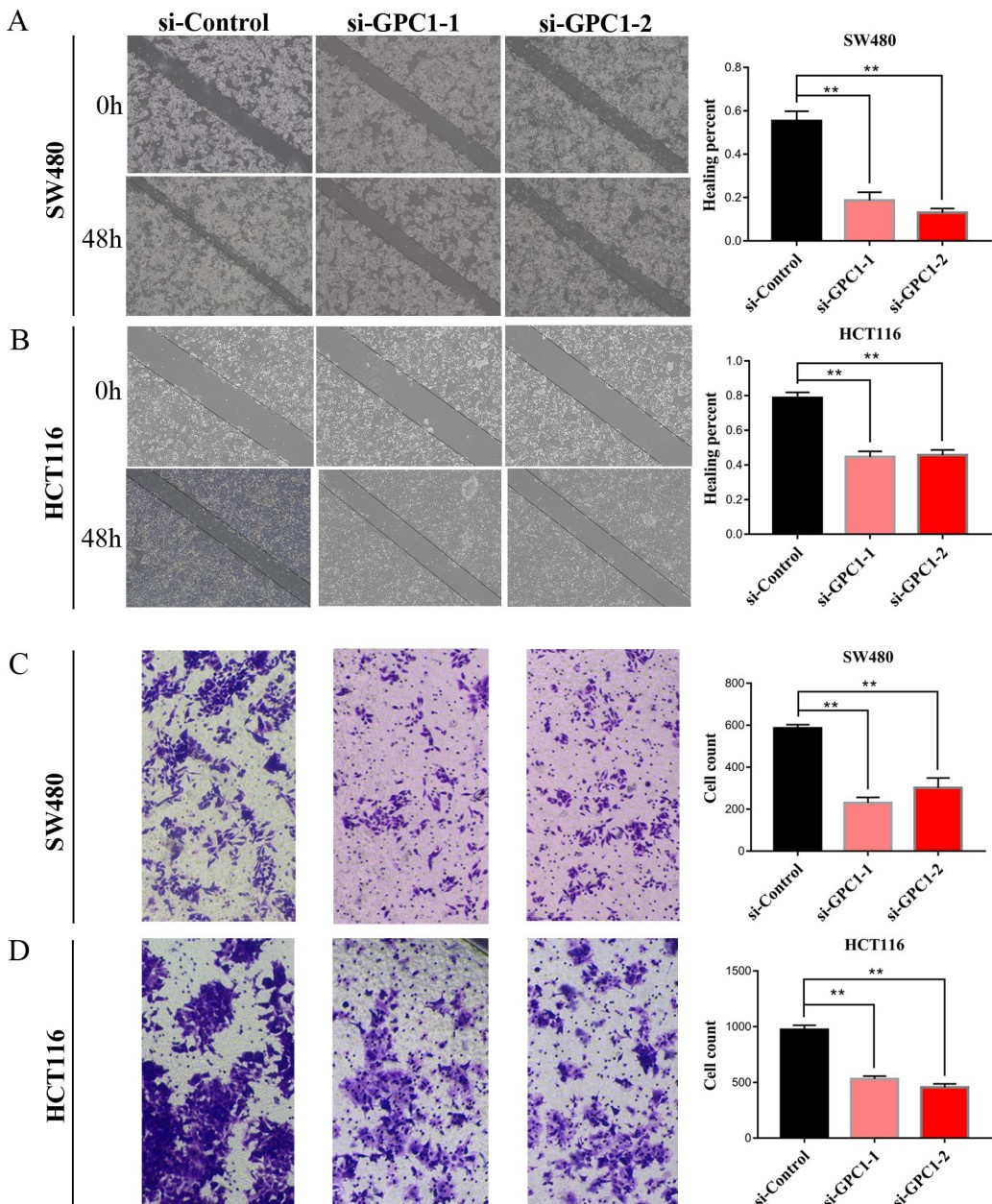

**Fig 5. Silencing GPC1 significantly reduces cell migration ability in CRC cells.** (A, B). Knockdown of GPC1 significantly inhibited wound healing in SW480 and HCT116 cell lines. (C, D). Silencing GPC1 significantly impaired the cell migration in SW480 and HCT116 cell lines. ** $p < 0.01$.

GPC3 also promotes the growth of liver cancer cells and may regulate the TGF-β signaling pathway [26]. The human TGF-β family includes 33 genes that encode for homodimeric or heterodimeric secreted cytokines [27, 28]. These proteins are involved in a variety of biological processes, including proliferation, differentiation, migration, apoptosis and adhesion [29]. Dysregulation of TGF-β signaling frequently occurs in CRC [30]. TGF-β signaling is initiated through binding of the TGF-β ligand to the receptor, which in turn phosphorylates the downstream target SMAD2/3 [31, 32]. Phosphorylated SMAD2/3 enters the nucleus with the help of SMAD4 and cofactors and acts as a transcription factor to

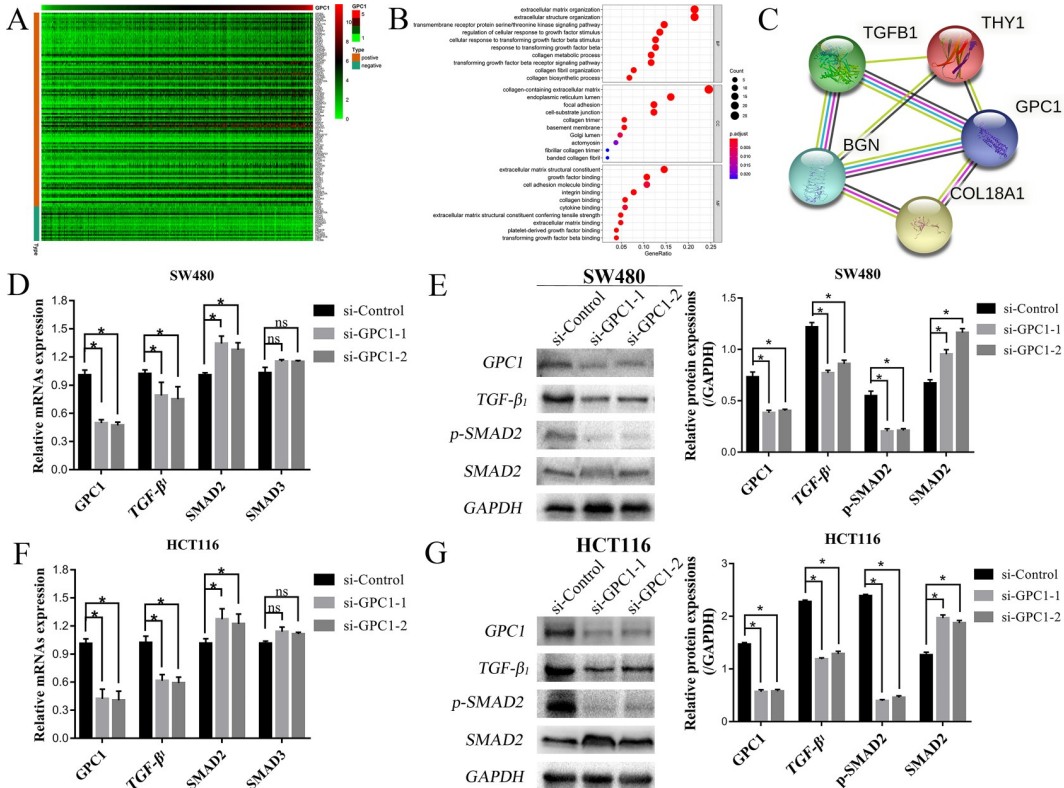

**Fig 6. Pathway enrichment analysis and TGF-β signaling pathway identification in CRC.** (A). Correlations between GPC1 and target genes in CRC. (B). GO analysis showed that the TGF-β signaling pathway was enriched in CRC. (C). The correlation-target network for GPC1. (D–G). GPC1-associated mRNAs and proteins were examined by qRT-PCR and western blot analysis in SW480 and HCT116 cell lines after silencing GPC1.

induce gene expression, including the expression of EMT-related genes [33, 34]. EMT has been shown to affect cancer cell migration [31]. The TGF-β/SMAD2/3 signaling pathway has been shown to affect the cell proliferation, survival, differentiation, apoptosis and migration of CRC [35–37]. The mechanism of S phase arrest and apoptosis induced by GPC1 silencing requires further investigation.

Future studies should verify our results using larger numbers of clinical specimen and more in-depth research is required into the mechanism by which GPC1 functions in CRC. In addition, animal models will be required to elucidate how GPC1 affects the occurrence and development of CRC in vivo. Our results suggest that GPC1 may be a prognostic marker and a new therapeutic target for CRC.

## Supporting information

**S1 Fig. GPC1 mRNA expression in different cancers and cancer cell lines.** (A) Significantly high expression of GPC1 mRNA in colon adenocarcinoma (COAD), breast invasive carcinoma (BRCA), cholangiocarcinoma (CHOL), esophageal carcinoma (ESCA), head and neck squamous cell carcinoma (HNSC), kidney renal papillary cell carcinoma (KIRP), lung squamous cell carcinoma (LUSC), skin cutaneous melanoma (SKCM), stomach adenocarcinoma (STAD). (B) GPC1 mRNA expression levels in different cancer cell lines.
(TIF)

**S2 Fig. Enrichment analysis of GPC1 in major cancer-related signaling pathways.** GPC1 is significantly enriched in the TGF-β signaling pathway.
(TIF)

**S1 Table. GPC1 is significantly related to M stage in colorectal cancer patients from TCGA.**
(DOCX)

**S2 Table. Results of 3 independent replicate experiments of cell apoptosis.** Note: Survival analysis was performed by Kaplan–Meier test, and correlation analysis of clinicopathological characteristics was performed by Kolmogorov-Smirnov test; the numbers in the table represent the P value of the correlation analysis.
(DOCX)

**S3 Table. Results of 3 independent replicate experiments of cell cycle.**
(DOCX)

**S1 File.**
(DOCX)

## Acknowledgments

We thank Gabrielle White Wolf, PhD, from Liwen Bianji (Edanz) (www.liwenbianji.cn/ac), for editing the English text of a draft of this manuscript.

## Author Contributions

**Conceptualization:** Fei Lu, Huazhang Wu, Mulin Liu.

**Data curation:** Fei Lu, Shuran Chen, Xu Su.

**Formal analysis:** Weijun Shi.

**Funding acquisition:** Fei Lu, Huazhang Wu, Mulin Liu.

**Investigation:** Fei Lu, Shuran Chen, Xu Su.

**Methodology:** Fei Lu.

**Project administration:** Fei Lu, Mulin Liu.

**Resources:** Fei Lu.

**Validation:** Fei Lu, Shuran Chen, Xu Su.

**Visualization:** Fei Lu.

**Writing – original draft:** Fei Lu, Weijun Shi, Huazhang Wu, Mulin Liu.

**Writing – review & editing:** Fei Lu, Weijun Shi, Huazhang Wu, Mulin Liu.

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
