## [Decision Letter · Decision Letter 0]

8 Mar 2022

PONE-D-21-26390

GPC1 promotes the growth and migration of colorectal cancer cells through regulating the TGF-β1/SMAD2 signaling pathway

PLOS ONE

Dear Dr. Liu,

Thank you for submitting your manuscript to PLOS ONE. After careful consideration, we feel that it has merit but does not fully meet PLOS ONE’s publication criteria as it currently stands. Therefore, we invite you to submit a revised version of the manuscript that addresses the points raised during the review process.

Please make sure to submit a point-by-point reply to address all the concerns raised by the reviewers.

We look forward to receiving your revised manuscript.

Kind regards,

Hamidreza Montazeri Aliabadi

Academic Editor

PLOS ONE

Journal Requirements:

Reviewers' comments:

Reviewer's Responses to Questions

**Comments to the Author**

1. Is the manuscript technically sound, and do the data support the conclusions?

Reviewer #1: Yes

Reviewer #2: Yes

Reviewer #3: Yes

2. Has the statistical analysis been performed appropriately and rigorously? 

Reviewer #1: Yes

Reviewer #2: Yes

Reviewer #3: Yes

3. Have the authors made all data underlying the findings in their manuscript fully available?

Reviewer #1: Yes

Reviewer #2: Yes

Reviewer #3: Yes

4. Is the manuscript presented in an intelligible fashion and written in standard English?

Reviewer #1: No

Reviewer #2: Yes

Reviewer #3: Yes

5. Review Comments to the Author

Reviewer #1: There are several questions needed to be discussed.

1. The manuscript has to be polished by professional language polishing companies because there are a lot of typographical or grammatical errors.

2. TGF-β signaling is initiated through binding of the TGF-β ligand to the receptor, which in turn phosphorylates the downstream target SMAD2/3. In this study, authors reported that GPC1 knockdown decreased the expression of TGF-β and p-SMAD2. However, why total SMAD2 expression also be affected?

3. In figure 2k, the GAPDH expression obviously differs in CRC cells and NCM460. therefore, please provide more accurate figures.

4. Please discuss how GPC1 influences the expression of TGF-β? It functions as a transcript factor?

Reviewer #2: PONE-D-21-26390.

GPC1 promotes the growth and migration of colorectal cancer cells through regulating the TGF-β1/SMAD2 signalling pathway

The authors investigate the potential role of GPC-1 in colorectal cancer using in silico and in vitro techniques. While the role of GPC-1 has been well described in other cancers such as pancreatic and prostate, its role in colorectal cancer is poorly understood.

While there has been some published data on GPC-1 mRNA expression in colorectal vs normal tissues, and in GPC-1 exosomes in colorectal cancer, there has been limited investigation of the mechanisms by which GPC-1 may play a role in the biology of CRC.

The authors demonstrate that knockdown of GPC-1 in colon cancer cell lines reduces proliferation and migration, consistent with what has been observed in other cancer types. They have identified the TGF-b1/SMAD2 signalling axis as a potential mediator of GPC-1 function in CRC.

A major challenge in determining the expression of GPC-1 in different cancers is that expression of the mRNA should be correlated to the expression of the GPC-1 protein using immunohistochemistry. If tissue availability is an issue for this, the authors should comment on the likelihood that the relative expression identified by mRNA is reflected in the protein expression in tumours.

Similarly, the detection of GPC-1 via western blot in this paper is problematic. GPC-1 is known to be heparan sulphated. While the core protein of ~55kDa can be detected without lysate treatment, use of Heparanase can result in much higher levels of GPC-1 detectable via western blot. The western blots shown in the figures and in the supplementary material do not have a positive control (e.g. recombinant GPC-1) to ensure that the band detected is actually GPC-1 (it is not clear what the molecular weight of the reactive bands are).

Specific comments:

Line 77 – reference 9 (Quach) does not show the expression in endometrial, lung or other tumours. Suggest referring to individual papers for each indication or a review article.

Line 177 – Western blot. GPC-1 is generally highly glycosylated – in particular with heparan sulfate chains. Was Heparanase treatment of samples attempted? If so was there any additional GPC-1 that became visible after treatment?

Line 186 – all glypicans seem to be differentially expressed at a statistically significant level between normal and tumour tissue. It appears GPC-3 and GPC6 are decreased, while GPC-4 is increased in tumour vs normal. It is not possible to tell from Figure 1 what the expression levels of GPC-5 are like in tumour vs normal.

Lines 192-200. While the differences in GPC-1 expression for tumour stage are statistically significant, the relative fold changes are small.

Line 196 – referring to Figure 2 western blots. The western blot for GPC-1 is very dark and the signal quite weak. The fold change in expression between normal and cancer lines shows error bars but it is unclear how many times these blots were performed to generate the densitometry. There is also no control for GPC-1 (e.g. recombinant GPC-1 or a cell line known to overexpress GPC-1).

Under the Methods section line 169 it is unclear if NCM460, RKO, CACO2 cells were also lysed in RIPA buffer and their growth conditions are not described in the Methods section.

Line 219 – Figure 3. GPC1 mRNA expression levels were knocked down by up to 60% but not 100%. This should be commented on. It is also not clear from the text or the figure legends what si-GPC-1-1 and si-GPC1-2 refer to (although this is mentioned in the methods). For 3B and D, it is not clear when the transient transfection of the siRNAs occurred relative to the start of the proliferation assay.

There is insufficient detail in Apoptosis and Cell Cycle methods to describe Fig 4. Also no information about number of times the experiment was repeated to generate the error bars is provided.

Figure 4 shows relatively minor changes in Apoptosis and Cell Cycle effects, although they appear from the data to be statistically significant.

Line 233 – cell migration. Again it is unclear when the GPC-1 knockdown was performed relative to the assay being performed.

Line 253 seems to have missed completion of a sentence.

Line 260 – protein levels of TGFb1. Presumably these are cellular levels as secreted TGFb1 would not be collected and analysed via the western blot protein extraction method described?

Line 271 – “after treatment with” rather then “after treated with”

Reviewer #3: Summary

This paper introduces the promoting role of GPC1 in CRC, and provides evidence that GPC1 may be used as an early biological detection standard and therapeutic target for CRC in the future, which can help doctors better treat CRC patients and has very high clinical significance.

Advantages

1、With clear logic and extensive data, this paper provides sufficient evidence that GPC1 promotes the development of CRC

2、In this paper, the functional pathway of GPC1 has also been confirmed.

Disadvantages

1、The picture is not clear enough. For example, the legend in the survival analysis of B in figure1 is not clear.

2、The expression levels of GAPDH in Western blot were significantly different, such as K in figure2

3、Since there are many figures in this paper, each figure can be placed under the corresponding results for easy reading

4、The language needs to be more refined

6. PLOS authors have the option to publish the peer review history of their article (what does this mean?). If published, this will include your full peer review and any attached files.

Reviewer #1: No

Reviewer #2: No

Reviewer #3: No

---

## [Author Response · Author response to Decision Letter 0]

30 Apr 2022

Reviewer Comments:

Reviewer 1

Question 1: The manuscript has to be polished by professional language polishing companies because there are a lot of typographical or grammatical errors.

Response: Thank you for your suggestions, our manuscript has been polished by Liwen Bianji (Edanz) (www.liwenbianji.cn/ac) for professional language polishing.

Question 2: TGF-β signaling is initiated through binding of the TGF-β ligand to the receptor, which in turn phosphorylates the downstream target SMAD2/3. In this study, authors reported that GPC1 knockdown decreased the expression of TGF-β and p-SMAD2. However, why total SMAD2 expression also be affected?

Response: Thanks for this comment. In the case of our study, we used RIPA Lysis Buffer (50mM Tris(pH 7.4), 150mM NaCl, 1% NP-40，0.5% sodium deoxycholate, 

0.1% SDS, sodium orthovanadate, sodium fluoride, EDTA, leupeptin) to extract proteins, in which the nucleoproteins are difficult to extract. Our speculation is that SMAD2 nuclear translation affects total SMAD2. (M. Kretzschmar, et al. A mechanism of repression of TGFbeta/ Smad signaling by oncogenic Ras Genes Dev., 13 (7) (1999), pp. 804-816; M. Kretzschmar, et al. The TGF-beta family mediator Smad1 is phosphorylated directly and activated functionally by the BMP receptor kinase Genes Dev., 11 (8) (1997), pp. 984-995).

Question 3: In figure 2k, the GAPDH expression obviously differs in CRC cells and NCM460. therefore, please provide more accurate figures.

Response: Thanks for this comment. Because NCM460 cells and Caco-2 cells are difficult to grow, the total GPC-1 expression is low. Due to experimental techniques, the GAPDH expression obviously differs in CRC cells and NCM460. For the accuracy of the conclusion, we used IHC technique for clinical tissue samples to investigate the protein expression levels of GPC1 in colorectal cancer.

Question 4: Please discuss how GPC1 influences the expression of TGF-β? It functions as a transcript factor?

Response: Thank you. Decreased GPC1 expression suppresses pancreatic cancer cell growth by modifying TGF-β signaling. GPC1 has been shown to interact with TGF-β and its receptors to stabilize their assembly for enhanced Smad signaling. Downregulation of GPC1 expression resulted in a slightly altered response toward TGF-β1, activin-A, and BMP2 in terms of growth, p21 induction, and Smad2 phosphorylation, ultimately leading to decreased anchorage-independent growth of T3M4 and PANC-1 cells. 

Reviewer 2

Question 1: Line 77 – reference 9 (Quach) does not show the expression in endometrial, lung or other tumours. Suggest referring to individual papers for each indication or a review article.

Response: Thanks for this comment.We have referred to individual papers for each indication or a review article (line 68).

Question 2: Line 177 – Western blot. GPC-1 is generally highly glycosylated – in particular with heparan sulfate chains. Was Heparanase treatment of samples attempted? If so was there any additional GPC-1 that became visible after treatment?

Response: Thanks for this comment. Your suggestion is very correct. But we did not treat protein samples with Heparinase. (Yu Mu, Dezhi Wang, Liangyu Bie,et al. Glypican-1-targeted and gemcitabine-loaded liposomes enhance tumor-suppressing effect on pancreatic cancer. Aging (Albany NY). 2020 Oct 15; 12(19): 19585–19596. ) 

Question 3: Line 186 – all glypicans seem to be differentially expressed at a statistically significant level between normal and tumour tissue. It appears GPC-3 and GPC6 are decreased, while GPC-4 is increased in tumour vs normal. It is not possible to tell from Figure 1 what the expression levels of GPC-5 are like in tumour vs normal.

Response: Thanks for your comment. We have produced a new Figure showing the significant differential expression of GPC5 between tumor and normal tissues.

Question 4: Lines 192-200. While the differences in GPC-1 expression for tumour stage are statistically significant, the relative fold changes are small.

Response: Thanks for this comment. Our data and scientific conclusion from them (FC>1 or FC <-1, P<0.05) are to be credible. (Shi, L., Jones, W. D., Jensen, R. V., Harris, S. C., Perkins, R. G., Goodsaid, F. M., … Tong, W. (2008). The balance of reproducibility, sensitivity, and specificity of lists of differentially expressed genes in microarray studies. BMC Bioinformatics, 9(Suppl 9), S10.)

Question 5: Line 196 – referring to Figure 2 western blots. The western blot for GPC-1 is very dark and the signal quite weak. The fold change in expression between normal and cancer lines shows error bars but it is unclear how many times these blots were performed to generate the densitometry. There is also no control for GPC-1 (e.g. recombinant GPC-1 or a cell line known to overexpress GPC-1).

Response: Thanks for this comment. Because Caco-2 cells are difficult to grow, the total GPC-1 expression is low. For the accuracy of the conclusion, we used IHC technique for clinical tissue samples to investigate the protein expression levels of GPC1 in colorectal cancer.

Question 6: Under the Methods section line 169 it is unclear if NCM460, RKO, CACO2 cells were also lysed in RIPA buffer and their growth conditions are not described in the Methods section.

Response: Thanks for this comment. SW480, NCM460, RKO, CACO2 and HCT116 cells were lysed using RIPA buffer (Beyotime, Shanghai, China). 

Question 7: Line 219 – Figure 3. GPC1 mRNA expression levels were knocked down by up to 60% but not 100%. This should be commented on. It is also not clear from the text or the figure legends what si-GPC1-1 and si-GPC1-2 refer to (although this is mentioned in the methods). For 3B and D, it is not clear when the transient transfection of the siRNAs occurred relative to the start of the proliferation assay.

Response: Thanks for this comment. On days 0 (at 48 h after transfection), 1, 2 and 3, CCK-8 reagent (10 µL) was added to cells and cells were incubated for 4–6 h.

Question 8: There is insufficient detail in Apoptosis and Cell Cycle methods to describe Fig 4. Also no information about number of times the experiment was repeated to generate the error bars is provided.Figure 4 shows relatively minor changes in Apoptosis and Cell Cycle effects, although they appear from the data to be statistically significant.

Response: Thanks for this comment. Concerning the under-description of Figure 4 in the Apoptosis and Cell Cycle Methods in this manuscript, we have described it exhaustively at the corresponding location in the manuscript, and the information about number of times the experiment was repeated to generate the error bars is provided in Supplementary Data. For the relatively small changes in apoptosis and cell cycle effects in Figure 4, because of their statistical differences and good statistical significance in other experimental stages, it does not affect the final conclusion of this study, so they are not further optimized.

Question 9: Line 233 – cell migration. Again it is unclear when the GPC-1 knockdown was performed relative to the assay being performed.

Response: Thanks for this comment. At 48 h after transfection, a sterilized 10-µL pipette tip was used to create a scratch in the monolayer of transfected CRC cells cultured in 6-well plates.

Question 10: Line 253 seems to have missed completion of a sentence.

Response: Thank you. Two different bioinformatics analyses indicated that GPC1 may regulate the TGF-β1 signaling pathway in CRC cells (Fig. 6B and 6C), which is consistent with our previous findings. 

Question 11: Line 260 – protein levels of TGFb1. Presumably these are cellular levels as secreted TGFb1 would not be collected and analysed via the western blot protein extraction method described?

Response: We thank the reviewer for the very interesting comment. In fact, we used RIPA Lysis Buffer to extract proteins, in which secreted TGFb1 and the nucleoproteins are difficult to extract. So we described how to extract the protein in the protein extraction method.

Question 12: Line 271 – “after treatment with” rather then “after treated with”

Response: Thank you. GPC1-associated mRNAs and proteins were examined by qRT-PCR and western blot analysis in SW480 and HCT116 cell lines after silencing GPC1.

Reviewer 3

Question 1: The picture is not clear enough. For example, the legend in the survival analysis of B in figure1 is not clear.

Response: Thank you. We made a new figure1.

Question 2: The expression levels of GAPDH in Western blot were significantly different, such as K in figure2.

Response: Thanks for this comment. Because NCM460 cells and Caco-2 cells are difficult to grow, the total GPC-1 expression is low. Due to experimental techniques, the GAPDH expression obviously differs in CRC cells and NCM460. For the accuracy of the conclusion, we used IHC technique for clinical tissue samples to investigate the protein expression levels of GPC1 in colorectal cancer.

Question 3: Since there are many figures in this paper, each figure can be placed under the corresponding results for easy reading.

Response: Thank you. We will place figures as required by the journal.

Question 4: The language needs to be more refined.

Response: Thank you for your suggestions, our manuscript has been polished by Liwen Bianji (Edanz) (www.liwenbianji.cn/ac) for professional language polishing.

---

## [Decision Letter · Decision Letter 1]

16 May 2022

GPC1 promotes the growth and migration of colorectal cancer cells through regulating the TGF-β1/SMAD2 signaling pathway

PONE-D-21-26390R1

Dear Dr. Liu,

We’re pleased to inform you that your manuscript has been judged scientifically suitable for publication and will be formally accepted for publication once it meets all outstanding technical requirements.

Kind regards,

Hamidreza Montazeri Aliabadi

Academic Editor

PLOS ONE

Additional Editor Comments (optional):

Reviewers' comments:

Reviewer's Responses to Questions

**Comments to the Author**

1. If the authors have adequately addressed your comments raised in a previous round of review and you feel that this manuscript is now acceptable for publication, you may indicate that here to bypass the “Comments to the Author” section, enter your conflict of interest statement in the “Confidential to Editor” section, and submit your "Accept" recommendation.

Reviewer #1: (No Response)

Reviewer #2: All comments have been addressed

Reviewer #3: All comments have been addressed

2. Is the manuscript technically sound, and do the data support the conclusions?

Reviewer #1: (No Response)

Reviewer #2: Yes

Reviewer #3: Yes

3. Has the statistical analysis been performed appropriately and rigorously? 

Reviewer #1: (No Response)

Reviewer #2: Yes

Reviewer #3: Yes

4. Have the authors made all data underlying the findings in their manuscript fully available?

Reviewer #1: (No Response)

Reviewer #2: Yes

Reviewer #3: Yes

5. Is the manuscript presented in an intelligible fashion and written in standard English?

Reviewer #1: (No Response)

Reviewer #2: Yes

Reviewer #3: Yes

6. Review Comments to the Author

Reviewer #1: (No Response)

Reviewer #2: (No Response)

Reviewer #3: After a rigorous revison on this manuscript, the manuscript is suitable for pubulication with sound quality on science.

7. PLOS authors have the option to publish the peer review history of their article (what does this mean?). If published, this will include your full peer review and any attached files.

Reviewer #1: No

Reviewer #2: No

Reviewer #3: No

---

## [Editor Report · Acceptance letter]

27 May 2022

PONE-D-21-26390R1 

GPC1 promotes the growth and migration of colorectal cancer cells through regulating the TGF-β1/SMAD2 signaling pathway 

Dear Dr. Liu:

I'm pleased to inform you that your manuscript has been deemed suitable for publication in PLOS ONE. Congratulations! Your manuscript is now with our production department. 

Kind regards, 

on behalf of

Dr. Hamidreza Montazeri Aliabadi 

Academic Editor

PLOS ONE